# OpenReview forum: "Litmus Tests—Quantifying How LLMs Trade Off Competing Objectives: Economic Decisionmaking as a Case Study"
_ICLR.cc/2026/Conference — Submitted to ICLR 2026_

### Official Review · Reviewer_558x · 2025-10-25

**Soundness:** 2
**Presentation:** 2
**Contribution:** 1
**Rating:** 2
**Confidence:** 5

**Summary:**

This paper proposes three different litmus tests for LLMs: Patience versus Impatience, Efficiency versus Equality, and Collusiveness versus Competitiveness. In contrast with benchmarks, litmus tests don’t encode a sense of optimality, but explore how LLMs deal with certain tradeoffs that can be encountered in everyday interactions. The results, with various SOTA models, show a range of values and character in different providers.

**Strengths:**

- Studying the behavior of models from the perspective of these tradeoffs, where there is not necessarily a metric to optimize, is really important. In the end, a lot of these qualities don’t have a clear optimum, but still have to be understood.
- The presentation is clear and easy to follow.

**Weaknesses:**

While I think this paper explores an interesting topic, I don’t believe it has the necessary breadth and/or depth.

There are many possible litmus tests, as you acknowledge, but you just implement three quite basic ones. I understand that it’s unreasonable to expect a lot more tests, but these are toy problems and are explored superficially. I think this work would benefit from creating realistic litmus tests to see the real impact of these tradeoffs. This seems particularly important to me because of the nature of these tests. There is no clear metric to optimize, so we can’t learn much from these in a vacuum.

I also don’t quite understand the rationale for using these litmus tests instead of others. Why did you think these were the most relevant at this point? Are these topics particularly important? I don’t see a good justification for using these instead of the others you briefly mentioned.

You discuss the idea of litmus tests like a new paradigm, but other papers establish tests (not benchmarks) that explore tradeoffs without a clear definition of correctness. For example, [1] explores the honesty and helpfulness trade-off in a lot more depth and detail. I encourage you to explore this literature a bit more, beyond Goli & Singh 2024 and Ross et al. 2024.

[1] Liu, R., Sumers, T. R., Dasgupta, I., & Griffiths, T. L. (2024). How do large language models navigate conflicts between honesty and helpfulness?. arXiv preprint arXiv:2402.07282.

Another weakness I see is the lack of a human baseline, considering how simple it is to collect one. You mention “our focus is on between-LLM comparison rather than on benchmarking how close LLMs are to humans,” but in this case, I think it’s crucial. Your litmus tests are by definition uncertain, so having a human baseline is an important reference point. Looking at the results, without the human baseline, makes it hard to reach any conclusions. Is there a model that is particularly better than others? Should we measure just one task or their ability to adapt to different ones? I’d even go further and say that you not only need a human baseline, but a way to see how these factors affect people interacting with these models. That’d allow us to better understand what to expect from interactions, and what we should strive for.

Finally, I don’t want to weigh this too much, but it’s a bit strange to submit partial results. For reference:

> For cost reasons and unexpected funding changes, we were unable to complete some of the runs of our most expensive litmus test—Collusiveness vs. Competitiveness (indicated by “—”). We will make every effort to collect this data by the rebuttal period.

**Questions:**

See weaknesses

---

> ### Author Response · Authors · 2025-12-02
>
> > I think this work would benefit from creating realistic litmus tests to see the real impact of these tradeoffs. This seems particularly important to me because of the nature of these tests. There is no clear metric to optimize, so we can’t learn much from these in a vacuum.
>
> Many tradeoffs can’t be cleanly characterized, which makes quantifying tradeoff responses tricky (what behavior corresponds to 0? to 1? how to interpolate between? can it be done in a cheap and trustworthy way, rather than having to rely on expensive human labels or unreliable LLM judge labels?). To address this issue, with our litmus tests, we focus on choice behavior of LLMs in **grounded** economic domains. Because the domains are grounded, there *are* in fact natural metrics to consider. For example, in Efficiency versus Equality, we can directly compute the degree of efficiency (company revenue) or equality (equality in worker pay) from any allocation the LLM proposes, using simple mathematical formulas given in the paper.
>
> > You discuss the idea of litmus tests like a new paradigm, but other papers establish tests (not benchmarks) that explore tradeoffs without a clear definition of correctness. For example, [1] explores the honesty and helpfulness trade-off in a lot more depth and detail.
>
> We focus on tradeoffs in grounded domains in which an agent’s actions can be quantified without ambiguity. An unavoidable disadvantage of our approach is that our environments appear more stylized, but an advantage is the lack of ambiguity in measurement (and therefore, as mentioned above, lack of reliance on expensive human labels or unreliable LLM judge labels). We think both [1] and our litmus test approach hold merit, and with this work, aim to build out the litmus test framework as a proof of concept.
>
> > Your litmus tests are by definition uncertain, so having a human baseline is an important reference point.
>
> Our litmus tests tests are by definition *not* uncertain -- our contribution lies precisely in designing controlled environments in which we can quantify tradeoff responses without ambiguity.
>
> > Looking at the results, without the human baseline, makes it hard to reach any conclusions. Is there a model that is particularly better than others?
>
> The purpose of litmus tests is to measure tradeoff responses. Thus, there is no notion of “better” or worse performance.

---

### Official Review · Reviewer_vWhw · 2025-11-01

**Soundness:** 2
**Presentation:** 1
**Contribution:** 2
**Rating:** 2
**Confidence:** 4

**Summary:**

- The paper introduces "litmus tests," a new type of LLM eval. Unlike benchmarks that measure raw capability, litmus tests quantify behavioral tendencies when faced with real-world decisions that involve tradeoffs and have no single correct answer.
- The authors design and test three litmus tests based on fundamental economic tradeoffs: (1) Patience vs. Impatience: A single-shot test that measures an LLM's intertemporal choice; (2) Efficiency vs. Equality: A repeated-interaction test where an LLM trades off maximizing total company revenue (efficiency) with ensuring workers receive equal pay (equality); (3) Collusiveness vs. Competitiveness: A multi-agent test where two competing LLMs collude (setting high prices for joint profit) or compete (setting low, Nash equilibrium prices).
- The paper evaluates frontier LLMs and find that they differ w.r.t. these axes.

**Strengths:**

The idea of a class of litmus tests is a somewhat novel idea, even if it does subsume some experiments that have previously been made in the literature. Overall, I don't think the contributions are particularly notable.

**Weaknesses:**

- The paper is notably difficult to read because it contains no figures, charts, or graphs. The main results are presented in Table 1, which is just a dense list of numbers.
- The paper's own methodology, the "reliability score", forces the authors to discard results from several modern LLMs. Gemini 1.5 Pro, for example, is excluded from the entire discussion because it failed the reliability check on all three tests. This suggests the proposed "litmus test" framework is not robust enough to evaluate all models.
- The paper makes strong claims about measuring "character" and "values". However, it only presents three tests, all confined to the domain of economics. It provides no evidence that an LLM's "patience" in a financial interest rate problem is a stable, generalizable trait.
- It is unclear what the broader relevance of these findings are; they seem narrow in their significance.

**Questions:**

None.

---

> ### Author Response · Authors · 2025-12-02
>
> Weakness 1: The reviewer is requesting visualizations of the data in Table 1. We are happy to include these.
>
> Weakness 2: The reviewer writes: “The paper's own methodology, the "reliability score", forces the authors to discard results from several modern LLMs.” This is a feature, not a bug. The purpose of the reliability score is to understand whether we can interpret an LLM’s choice behavior as a coherent “preference”. This point is also already explained in Ln 128-130 of the paper. Moreover, Table 1 indicates that more recently released LLMs tend to have higher reliability scores, indicating that this “problem” of insufficient reliability is likely to diminish as LLM capabilities improve.
>
> Weakness 3: Regarding generalizability, we already address this directly in Ln 73-81, in which we show that the results of our Efficiency vs. Equality litmus test closely track the results of Westwood et al. (2025)’s ModelSlant, an evaluation of political bias. Regarding there “only” being three tests -- our contribution is methodological in nature, and serves as a proof of concept to highlight the potential value of the litmus test paradigm.

---

### Official Review · Reviewer_7q2e · 2025-11-01

**Soundness:** 3
**Presentation:** 2
**Contribution:** 2
**Rating:** 2
**Confidence:** 4

**Summary:**

This paper introduces "litmus tests" as quantitative measures to evaluate LLMs' behavioral tendencies when facing decisions involving tradeoffs with no objectively correct answer. The authors develop specific litmus tests for three economic tradeoffs: efficiency versus equality, patience versus impatience, and collusiveness versus competitiveness. While the authors frame this as a novel contribution, similar approaches have been extensively explored in the psychological and cognitive testing literature on LLMs, where decision-making tendencies and preferences have been measured under various names. The primary distinction claimed by the authors is an increased emphasis on inter-LLM comparisons.

**Strengths:**

**1.** **Comprehensive experimental design**: The paper demonstrates how these tests can be applied in both single-shot and repeated settings, as well as across tasks of varying complexity levels, providing a thorough exploration of the proposed framework.

**2.** **Rigorous operationalization**: The authors develop carefully designed, specific measures for each economic tradeoff, with clear quantitative definitions that facilitate systematic comparison across models.

**3.** **Ecological validity**: The inclusion of tool use in economic tasks is an interesting design choice that enhances the ecological validity of the experiments and better reflects real-world decision-making scenarios where LLMs have access to computational resources.

**Weaknesses:**

**1.** **Limited novelty and unclear differentiation from existing work**: The concept of eliciting cognitive phenotypes or economic preferences from AI systems has substantial precedent in the literature, though perhaps under different terminologies. The paper does not clearly distinguish how "litmus tests" differ fundamentally from existing measurements of decision-making characteristics such as risk attitude, loss aversion, or temporal discounting rates—all of which involve competing demands and reflect preferences rather than capabilities. Similarly, the relationship to personality testing frameworks remains unclear. The authors acknowledge that many related preferences (exploration vs. exploitation, free riding vs. altruism) exist as future directions, yet many of these have already been investigated in psychological and cognitive studies of LLMs. The stated contribution of "increased emphasis on inter-LLM comparisons" (line 121) appears incremental unless generalizable principles emerge from such comparisons.

**2.** **Confounding factors and measurement validity**: When goals are not explicitly defined or understood by the model, it becomes difficult to determine what the test actually measures. The measurements are highly dependent on prompt phrasing, which introduces potential confounds. For instance, the efficiency vs. equality scenario is framed as a tradeoff between company efficiency and worker equality, but more precisely represents a tension between meritocracy (more able workers receiving higher pay) and egalitarianism (equal pay despite random task assignment). Such ambiguities raise concerns about the tests' transferability and predictive validity.

**3.** **Lack of mechanistic insight and interpretability**: The paper provides limited interpretability regarding why LLMs exhibit particular behavioral patterns. Without understanding the underlying mechanisms, these measurements merely transform one black box (the LLM) into another (the litmus score), offering little predictive power about how these scores will influence LLM behavior in novel contexts. It remains unclear whether the observed decisions reflect genuine internal value systems or are artifacts of prompting and training.

**4.** **Questionable generalizability and stability**: The paper does not demonstrate whether litmus scores represent stable characteristics consistent across different tasks and prompts. Without evidence of cross-task stability, the utility of these measures is limited. The ability to extract meaningful "character" traits (analogous to personality) depends on their effective transfer across tasks. For example, the authors interpret patience vs. impatience as reflecting different internal interest rates, but without stability guarantees, these measurements may not reliably predict behavior. If these are not stable characteristics, their practical value for understanding or predicting LLM behavior becomes questionable.

**5.** **Problematic measure definition conflating capacity and preference**: In the efficiency vs. equality test, the litmus score is defined relative to two baselines (maximizing efficiency and maximizing equality). However, because the task is complex and LLMs may not perfectly execute either strategy even without value conflicts, this approach conflates capability with preference in ways that are difficult to disentangle. This issue parallels a known artifact in human metacognition research, where initial findings suggesting correlation between perceptual capacity and metacognitive ability were later shown to be spurious. If an LLM cannot reliably achieve the required performance, its scores will inevitably fluctuate between the two extremes due to capacity limitations rather than genuine preference. Forcing scores between 0 and 1, as in the current manuscript, may mask rather than resolve this problem.

**Questions:**

### Q1: How do litmus tests differ from existing preference measures applied to LLMs?
What unique insights do litmus tests provide? Either (a) demonstrate empirically or theoretically that litmus tests capture phenomena not measurable by existing frameworks, or (b) reposition your work as applying established paradigms to LLMs, focusing on what generalizable principles emerge from systematic inter-model comparisons.

### Q2: How sensitive are litmus scores to prompt variations, and what exactly are the tests measuring?
For example, efficiency vs. equality could be interpreted as company efficiency vs. worker equality, or meritocracy vs. egalitarianism. What goals do LLMs perceive?

### Q3: Do litmus scores represent stable characteristics?
Without demonstrating cross-task consistency, these may be task-specific responses rather than stable "character" traits with predictive value. Validating that (a) scores remain consistent across different instantiations of the same tradeoff, and (b) scores predict behavior in novel, related tasks would be helpful.

### Q4: What drives the observed patterns, and do scores reflect genuine internal values?
Including interpretability analyses would help.

### Additional Minor Comments
* The term "litmus test" in the title may be overly jargon-heavy and could benefit from clarification or a more accessible framing.
* The notation in Section 3.2.1 is confusing, as task *i* is not necessarily assigned to worker *i*. Using distinct subscripts for tasks and workers would improve clarity.
* The presentation of experimental procedures would benefit from showing example prompts before summarizing the experimental design, as this would help readers understand the correspondence between prompt content and design choices, particularly for complex procedures like efficiency vs. equality. Clarification regarding how much LLMs understood about task structures would also be valuable.
* The rationale for implementing efficiency vs. equality as a repeated task should be explicitly stated.

---

> ### Author Response · Authors · 2025-12-02
>
> This review appears to be partly or fully AI-generated: There are factual inaccuracies and irrelevant points raised in this review that would not have been included if a human had carefully checked the contents.
>
> A main theme in the reviewer’s comments is that of measurement validity and generalizability (Weakness 2, 3, 4, 5) -- however, the issues the reviewer raises are ones that we already directly address in the paper:
> - The concern regarding generalizability is already addressed in the paper in Lines 73-81, in which we show that the results of our Efficiency vs. Equality litmus test closely track the results of Westwood et al. (2025)’s ModelSlant, an evaluation of political bias.
> - The concern regarding conflating capability with preference is already addressed. In fact, it is one of the central methodological contributions of the paper. Specifically, for each litmus test, we consider two metrics -- a reliability score, measuring capability, and a litmus score, measuring character/values/preference assuming sufficient capability.
>
> Other points raised by the reviewer include expanding related work, which is something we are happy to do, subject to space constraints.

---

### Official Review · Reviewer_qD5u · 2025-11-02

**Soundness:** 3
**Presentation:** 2
**Contribution:** 2
**Rating:** 2
**Confidence:** 3

**Summary:**

This paper introduces Litmus tests to evaluate LLMs’ character, value, and behavioral tendencies when faced with tradeoffs. A Litmus test considers an LLM’s decision-making with respect to two objectives and outputs a quantitative score measuring its trade-offs between the objectives.

This paper develops three Litmus tests: a single-shot Litmus test evaluating patience vs. impatience tradeoff, a repeated-interaction test evaluating efficiency vs. equality tradeoff, and a multi-agent test evaluating collusiveness vs. competitiveness tradeoff. The authors apply these tests on a broad range of LLMs and report findings about different LLMs’ behaviors and characters.

**Strengths:**

1.	The proposed Litmus tests output quantitative scores indicating trade-off behaviors that can be compared across LLMs.

2.	The proposed Litmus tests handle three representative trade-offs in different contexts: single-shot, repeated interaction and multi-agent environment.

**Weaknesses:**

1.	Unclear value in understanding LLM trade-off behavior: From the paper, I am unconvinced about the value in understanding LLM trade-off behaviors using a tool such as Litmus test. The paper concludes by stating that “We expect that litmus test scores that measure “LLM personality” will complement benchmark scores in aiding decisions about whether and which LLMs to deploy in various use cases”. This is a bold claim that needs to be thought through and elaborated further. An implicit assumption appears to be that LLMs have personality that consistently influence how they trade-off competing objectives. One may likely hold an opposite view that is LLMs do not have character or personality – we can use carefully crafted prompts or fine-tuning to attain specific trade-off behaviors from LLMs.

2.	Insufficient comparison with literature exploring LLM tradeoff responses: Section 2 mentions a few related works that have explored LLM tradeoffs in different domains. Direct and detailed comparison with literature would be helpful to position the paper among the growing set of works. Right now, it is difficult to see how much novelty and value Litmus tests are contributing. One possible way to connect more closely to literature is to consider some tradeoff domains studied in early works, and develop Litmus tests for these domains. This may allow direct comparison that clarify the contribution unique to Litmus tests.

3.	Lack general guidance on developing Litmus tests: The paper uses case studies to demonstrate possible designs of Litmus tests, but there lacks general guidance on applying this method. In all three case studies, the two “conflicting” objectives involved in trade-off can be characterized clearly, leading to well-defined Litmus score definition. What about cases where trade-offs are more complex, and neither of the “extreme cases” (corresponding to optimizing either objective) can be cleanly characterized? Some general guidance on defining Litmus score would be beneficial to include.

**Questions:**

Please refer to my comments and questions listed in weaknesses.

---

> ### Author Response · Authors · 2025-12-02
>
> Regarding Weakness 1: “One may likely hold an opposite view that is LLMs do not have character or personality – we can use carefully crafted prompts or fine-tuning to attain specific trade-off behaviors from LLMs.” The question this paper aims to address is precisely this one, on whether LLMs have some degree of inherent character or personality. We think our discussion comparing our Efficiency vs. Equality results to the external ModelSlant results presents some compelling evidence in this direction. We also remark it’s not a binary yes/no question, but rather a question of to what extent do LLMs have persistent character or personality in various dimensions.
>
> Regarding Weakness 2: The reviewer requests more comparison to related work. We’re happy to expand on this.
>
> Regarding Weakness 3: A key methodological insight we aim to raise with this paper, which we will clarify more, is the value of our testing tradeoff responses in **grounded domains**. As you mention, many tradeoffs can’t be cleanly characterized, which makes assigning litmus scores tricky (what behavior corresponds to 0? to 1? how to interpolate between? can it be done in a cheap and trustworthy way, rather than having to rely on expensive human labels or unreliable LLM judge labels?). This is why in our work we focus on economic tradeoffs like “Patience vs. Impatience” -- the tradeoff is simple and single-dimensional, so that LLM choice behavior has a clear economic interpretation.

---

### Meta-Review · Area_Chair_ifbM · 2026-01-07

**Summary:**

The paper topic is clearly quite interesting (as acknowledged by reviews, and I agree). The remaining concerns are largely about the domain-constrained, stylized tests and a variety of other more minor factors (noted below). Given the lack of an adequate rebuttal, it is difficult to support acceptance at this point.

**Reviewer Concerns:**

Several of the concerns were not addressed in the rebuttals. These include:
- **qD5u**: “general guidance on applying this method”
- **7q2e**’s concerns are largely waived away as due to LLM generation, but (1) this is difficult to verify, and (2) claiming this does not itself absolve the authors of the responsibility to write a clear response. The response: “expanding related work, which is something we are happy to do” does not address the concern about related work. Rather, it defers it. So, the response in all addresses W4 and W5 (albeit a bit vaguely), but does not address W1, W2, and W3.
- **vWhw**: As before, “We are happy to include these” is insufficient; ICLR allows revisions, and this seems like a trivial one. What was the blocker to including it with the rebuttal? Perhaps more importantly, this seems like a misinterpretation of the reviewer’s comment, which is a comment about readability, not a request for a plot. It is also a perfectly reasonable question of whether inconsistent preferences should invalidate a model’s scoring, given that human choosers often have inconsistent preferences too. The answer trivializes this concern.
- **558x**: the explained comparison to [1] is brief and superficial, but this is a fair question/concern. While toy problems can be useful, the causal story about the value of this work and its relationship to [1] need to be clearer. The issue of a human baseline is glossed over rather than addressed (again, a fair question).

**Reviewer Scores:**

Unfortunately, given the brief rebuttals and lack of revisions, it seems unlikely the reviewers had cause to significantly move their scores up.

---

### Decision · Program_Chairs · 2026-01-26

Reject